# Regulation of Fructose 1,6-Bisphosphatase in Procyclic Form *Trypanosoma brucei*

**DOI:** 10.3390/pathogens10050617

**Published:** 2021-05-18

**Authors:** Christina Wilkinson, Meredith T. Morris

**Affiliations:** Eukaryotic Pathogens Innovation Center, Department of Genetics and Biochemistry, Clemson University, Life Sciences Facility, 190 Collings Ave., Clemson, SC 29634, USA; cwilki7@g.clemson.edu

**Keywords:** fructose 1,6-bisphosphatase, *Trypanosoma brucei*, metabolism, gluconeogenesis, regulation

## Abstract

Glycolysis is well described in *Trypanosoma brucei*, while the importance of gluconeogenesis and one of the key enzymes in that pathway, fructose 1,6-bisphosphatase, is less understood. Using a sensitive and specific assay for FBPase, we demonstrate that FBPase activity in insect stage, procyclic form (PF), parasite changes with parasite cell line, extracellular glucose levels, and cell density. FBPase activity in log phase PF 2913 cells was highest in high glucose conditions, where gluconeogenesis is expected to be inactive, and was undetectable in low glucose, where gluconeogenesis is predicted to be active. This unexpected relationship between FBPase activity and extracellular glucose levels suggests that FBPase may not be exclusively involved in gluconeogenesis and may play an additional role in parasite metabolism. In stationary phase cells, the relationship between FBPase activity and extracellular glucose levels was reversed. Furthermore, we found that monomorphic PF 2913 cells had significantly higher FBPase levels than pleomorphic PF AnTat1.1 cells where the activity was undetectable except when cells were grown in standard SDM79 media, which is glucose-rich and commonly used to grow PF trypanosomes in vitro. Finally, we observed several conditions where FBPase activity changed while protein levels did not, suggesting that the enzyme may be regulated via post-translational modifications.

## 1. Introduction

*Trypanosoma brucei* is an extracellular parasite that causes human African trypanosomiasis (HAT) as well as the wasting disease, nagana, in cattle [1,2]. This neglected tropical disease is endemic to sub-Saharan Africa and it is estimated that 60–70 million people are at risk of infection [3].

*T**rypanosoma brucei* is transmitted through the bite of the tsetse fly to a mammalian host [4], undergoes dramatic environmental changes, and must remodel its metabolism to survive [5,6]. In the mammalian host, the parasite spends much of its lifecycle in the bloodstream, which contains glucose (5 mM). This abundant carbon source is critical for bloodstream form (BF) parasites as they rely almost exclusively on glycolysis for ATP production [7,8]. In the tsetse fly midgut, the procyclic form (PF) of the parasite primarily utilizes proline as a carbon source, generating ATP in their mitochondrion [5,9,10].

Glycolysis regulates the breakdown of glucose to generate ATP and gluconeogenesis mediates the synthesis of glucose from non-carbohydrate carbon sources. In most organisms, these two metabolic processes are localized to the cytosol and are coordinately regulated by the allosteric modulation of the glycolytic enzymes, hexokinase (HK) and phosphofructokinase (PFK) [11,12]. *Trypanosoma brucei* is unique in that most of the glycolytic and gluconeogenic pathways are localized to peroxisome-like organelles termed glycosomes. Furthermore, trypanosome homologs of HK and PFK are not responsive, or much less so, to effectors that mediate their activity in other systems [13].

While glycolysis is well studied in *Trypanosoma brucei*, the role of gluconeogenesis and a key enzyme in the pathway, fructose 1,6 bisphosphatase (FBPase), is less understood. Studies using BF crude lysates [14] and experiments with PF and BF lysates [15] failed to detect FBPase enzymatic activity, leading investigators to hypothesize that *Trypanosoma brucei* did not utilize gluconeogenesis. The identification of FBPase in the trypanosome genome [16] and the detection of metabolic intermediates of the glycolytic and pentose phosphate pathway in PF parasites grown in the absence of glucose [17] provided the first evidence that these parasites could utilize gluconeogenesis. Metabolic labeling experiments in BF trypanosomes did not detect gluconeogenesis or FBPase activity when cells were grown in media containing glucose as their primary carbon source [18]. However, BF parasites cultured in the presence of glycerol were able to utilize it to fuel gluconeogenesis when glucose uptake was inhibited [19]. In recent work, PF FBPase deletion mutants could not establish infection in tsetse flies, revealing the importance of gluconeogenesis and FBPase to PF parasites [20].

Historically, the enzymatic activity of FBPase has been difficult to monitor. Using a sensitive fluorescence-based assay, we detected FBPase activity in PF parasites and found that it was influenced by extracellular glucose levels in a cell density-dependent manner. Further, FBPase activity varied significantly between PF cell lines. Monomorphic PF 2913 cells had higher FBPase activity than pleomorphic PF AnTat1.1 cells. We also document cases in which FBPase activity levels differed while protein levels were unchanged, suggesting regulation via post-translational modification. These findings highlight the variation in metabolism and enzyme regulation between parasite strains and in response to changing environmental conditions. They also suggest a role for FBPase in parasite biology outside of its canonical role in gluconeogenesis.

## 2. Results

To measure FBPase activity, we optimized a coupled enzyme assay (Figure 1A) that uses fluorescence, which is more sensitive than absorbance at 340 nm, to measure NADPH production over time. Using this approach, we were able to detect FBPase activity with as little as 125 pg of recombinant enzyme (Figure 1B). Western blot analysis of whole cell lysates and serial dilutions of recombinant FBPase protein revealed that 5 × 10^6^ cells contained approximately 75 pg of FBPase (Figure 1C,D) and that this assay would likely detect activity, if present, in PF parasites.

Our assay was robust when used with glycosome-enriched fractions harvested from PF 2913 cells grown in SDM79, which is the standard PF culture media (Figure 2A). These glycosome-enriched fractions yielded activity that was linear throughout the reaction course, with a signal that was 200× above background (Figure 2B). No signal was detected when glycosome-enriched fractions were incubated alone or when PGI or G6PDH were eliminated from the reaction mixture (Appendix A). To confirm our assay was specific for FBPase, we measured activity in cell lines in which FBPase was silenced by RNAi (Figure 2C). FBPase activity in parental cells was approximately 600 RFU/min and undetectable in RNAi cells grown in tetracycline (Figure 2D). Confirming the impact of the silencing, FBPase protein was undetectable by Western blotting after RNAi.

Because the canonical role of FBPase is in gluconeogenesis under conditions of low glucose, we assessed the impact of extracellular glucose on FBPase activity. To modulate glucose in the media, we used SDM79Θ [21], which includes fetal bovine serum (FBS) that has been dialyzed to remove glucose. To account for the potential impact of the loss of critical small molecules other than glucose in the SDM79Θ, we used SDM79Θ supplemented with physiologically relevant levels of glucose (5 mM) to yield SDM79ΘG. We predicted that FBPase activity would be greater in the very low glucose medium (SDM79Θ) and reduced in SDM79ΘG. Surprisingly, cells grown in SDM79Θ exhibited minimal FBPase activity (~75 RFU/min) when compared to cells grown in SDM79ΘG (~200 RFU/min) (Figure 3A). Additionally, cells grown in SDM79, which contains undialyzed FBS and glucose, had significantly higher FBPase activity (~450 RFU/min) than cells cultured in either SDM79Θ or SDM79ΘG (Figure 3A).

Through the course of these studies, we noticed that FBPase activity changed with cell density. To score this more accurately, parasites were seeded at 5 × 10^5^ cells/mL and FBPase activity was measured at various densities until the cells reached stationary phase (1 × 10^7^ cells/mL). (Note: the experiments presented above were performed with cells in log (5 × 10^6^ cells/mL) phase.) In contrast to assays performed on log phase cells, where FBPase activity was highest in SDM79ΘG (~200 RFU/min), FBPase activity in stationary cells was highest in SDM79Θ (~150 RFU/min) and undetectable in SDM79ΘG (Figure 3B). Next, we compared FBPase activity levels in cells cultured in the different media at log and stationary growth phases. FBPase activity was density-dependent in SDM79Θ and SDM79ΘG. Cells grown in SDM79 had significantly higher FBPase activity than either of the other media. Additionally, activity in SDM79 was not density-dependent (Figure 3C). The above experiments indicated that FBPase activity is influenced by extracellular glucose levels and culture density.

To resolve whether differences in FBPase activity were due to changes in steady state protein abundance, we measured FBPase levels from log phase and stationary phase cultures grown in SDM79, SDM79Θ, or SDM79ΘG using Western blotting and densitometry. FBPase expression levels were highest in SDM79 as compared to SDM79Θ or SDM79ΘG (Figure 3D), with the latter two media yielding similar protein levels. Unlike FBPase enzyme activity, which changed with cell density, protein levels were similar in log and stationary phase cultures.

Growth of PF 2913 in SDM79Θ and SDM79ΘG was similar with doubling times ranging between 25 and 26 h. These cell lines grew faster in SDM79 with a doubling time of 20 h (Figure 3C).

The unexpected FBPase activity in PF 2913 may be a result of the natural history of this cell line. These parasites were engineered for tetracycline inducible gene expression of transgenes [22]. This cell line has become one of the standard strains for trypanosome genetic studies and has been cultured for decades in many laboratories, usually in SDM79 or similar media containing abundant glucose. PF 2913 cells are also monomorphic, meaning they cannot complete the parasite lifecycle. To address whether the unanticipated pattern of FBPase regulation was specific to PF 2913s, we measured FBPase activity in pleomorphic PF AnTat1.1 cells that can differentiate in culture and have no genetic modifications. Short stumpy parasites were isolated from infected rodents and then differentiated in culture in the absence of glucose. The resulting parasites have hallmarks of procyclic trypomastigotes expressing EP procyclin, while no longer expressing PAD1 [21]. The newly differentiated PF parasites were cultured in SDM79Θ, a condition that favors growth in these parasites [21]. To score their FBPase activity in response to culture conditions, cells were passed into either SDM79ΘG or SDM79. Because the presence of glucose initially inhibits PF AnTat1.1 cell growth, parasites were acclimated for two weeks (a period required for the parasites to resume growing in glucose-replete media [21]) before use.

After cells were acclimated to the different conditions, FBPase activity and steady-state protein abundance were scored. In SDM79, the PF AnTat1.1 cells had significant FBPase activity (~200 RFU/min) similar to PF 2913 parasites under the same condition (Figure 4A,B). However, FBPase activity was undetectable in PF AnTat1.1 cells cultured to log or stationary phase in either SDM79Θ or SDM79ΘG (Figure 5A,B). In SDM79, cell density did not impact FBPase activity (Figure 5C). As seen with PF 2913 cells, PF AnTat1.1 cells had significantly higher activity when grown in SDM79 as compared to either SDM79Θ or SDM79ΘG.

## 3. Discussion

In this work, we utilized a robust and sensitive assay to examine FBPase regulation in *Trypanosoma brucei*. We found that FBPase activity differed significantly between PF 2913 and PF AnTat1.1 cells and was influenced by extracellular glucose levels and cell density in a manner that was unanticipated. Lastly, we identified conditions in which FBPase activity levels changed while protein abundance remained constant, suggesting enzyme activity is modulated by post-translational modifications.

Previous studies [15] were unable to detect FBPase activity enzymatically using whole cell lysates from PF MITat1.1, formerly designated as 4278-12/ICI-060. Differences between our data and theirs could be due to differences in assay sensitivity, cell lines and culturing conditions. Our observation that these variables influence FBPase activity supports this hypothesis. Additionally, we perform our assays with glycosome-enriched fractions. Using whole cell lysates yielded data that were less reproducible. The unpredictable behavior was likely a result of a complex mixture of biochemical reactions that consume and generate NADPH.

Other indirect evidence suggests that FBPase activity is present in trypanosomes. Metabolic labeling studies revealed an increase in ^13^C incorporation into glycolytic and pentose phosphate pathways when labeled with [U-^13^C] proline in the glucose-depleted media, which could only be supported through gluconeogenesis [17]. In agreement with those labeling experiments, we detected FBPase enzyme activity in PF 2913 cells grown in low glucose.

Interestingly, we did not detect FBPase activity in PF AnTat1.1 cells in either SDM79ΘG or SDM79Θ media, conditions that did not alter parasite growth rates. In very low glucose conditions, we hypothesize that gluconeogenesis should be essential for the generation of glucose-6-phosphate (G6P). Therefore, it was surprising to us that we did not detect FBPase activity in parasites grown in SDM79Θ. It may be that FBPase activity was below the sensitivity of our assay but sufficient to maintain gluconeogenic flux. Alternatively, the cells may be generating G6P via another pathway or enzyme. The product of FBPase was detected in metabolic labeling experiments in FBPase deletion mutants, leading to the speculation that another enzyme, sedoheptulose 1,7-bisphosphatase, provides this essential activity [20]. Because we do not know the kinetic parameters or substrate specificities of SBPase, it is not clear if our assay would detect this activity.

We detected significant FBPase activity in PF 2913 but not in PF AnTat1.1 parasites grown in SDM79Θ and SDM79ΘG. These two strains have a very different history. The PF AnTat1.1 cells used in this study were recently generated from parasites passaged through a mammalian host via glucose depletion [21]. After differentiation, PF AnTat1.1 cells are routinely maintained in very low glucose SDM79Θ. These cells exhibit a growth deficiency when grown in high glucose media and lose their pleomorphic characteristics after continuous culturing in high glucose. In contrast, PF 2913 cells have been cultured for decades in high glucose media and glucose is not inhibitory [23]. It is likely that PF 2913 metabolism has changed during long-term culturing in high glucose conditions.

We were surprised that FBPase activity was undetectable in PFAnTat1.1 cells grown in SDM79Θ or SDM79ΘG, but present when cells were grown in SDM79, which contains high levels of glucose and undialyzed FBS. Furthermore, FBPase activity in PF 2913 cells was significantly higher in cells grown in SDM79 as compared to either SDM79Θ or SDM79ΘG. The observation that FBPase activity was always higher in SDM79 as opposed to SDM79ΘG suggests that a component of FBS may influence FBPase activity. This finding underscores the importance of undefined media components in influencing parasite metabolism. This is a fundamental issue in biology as it is difficult to (i) precisely define the environments parasites encounter in the field and (ii) recapitulate those conditions in the laboratory. It has been noted anecdotally that important small molecules that stimulate parasite growth are lost in the dialysis of FBS. While PF cells grow well in SDM79Θ and SDM79ΘG (Figure 3C and Figure 5C), it is impossible to discount the effect these small molecules may have on metabolism. It remains to be seen if FBS from different vendors or lots elicits this response.

To our knowledge, this is the only study that directly compares FBPase enzymatic activity in PF parasites grown in different growth conditions and we do not know if the increase observed in FBPase activity observed under some conditions is associated with higher rates of gluconeogenesis. The unexpected relationship between FBPase activity and glucose levels (higher FBPase activities in high glucose) suggests that there may be an extra-gluconeogenic function for the enzyme. Changes in FBPase activity that are not accompanied by changes in activity of other enzymes in the pathway would lead to an increase or decrease in levels of fructose-6-phosphate (F6P) and fructose 1,6 bisphosphate (FBP), two molecules that regulate other enzymes and processes. Because *Trypanosoma brucei* pyruvate kinase is allosterically activated by FBP [24], increasing FBPase activity in high glucose conditions might result in lower FBP levels and inhibition of pyruvate kinase. This could shuttle more flux to protein and nucleotide synthesis and away from glycolysis.

One function of FBPase may be to regulate levels of FBP in its capacity as a glucose sensor in the AMP-independent activation of AMPK upon glucose starvation. AMPK serves as a central regulator of metabolism and is responsive to cellular energy homeostasis. Recent studies suggest that FBP levels stabilize inactive AMPK through interactions that facilitate its phosphorylation [25]. In this situation, lower levels of FBPase activity would result in decreasing levels of FBP that would lead to more active AMPK. In this scenario, increasing FBPase activity could be a means of maintaining some active AMPK in high glucose conditions.

To our knowledge, this is the first documentation of a role for culture density on FBPase activity and protein expression. However, density-dependent processes are fundamental to the cellular biology of trypanosomes and other kinetoplastids. As BF density increases, long slender forms of the parasite perceive a quorum-dependent signal that triggers differentiation into short stumpy forms [26]. Parasite density also affects the social motility of *Trypanosoma brucei*. When PF AnTat1.1 parasites were plated at various concentrations, motility began only after the threshold of 1.5 × 10^6^ cells [27]. In the distantly related parasite, *Leishmania mexicana*, cell density and extracellular glucose levels were involved in the regulation of the glucose transporter LmxGT1 [28], which was downregulated with increasing cell density and upregulated when glucose was removed from the media.

The finding that FBPase activity in our assays changes while protein levels remain constant suggests the protein is post-translationally modified. In the plant *Pisum sativum*, the cytosolic FBPase 1 (*Ps*cFBP1) is redox activated by reduction of a disulfide bridge involving Cys153 and Cys173 [29]. The oxidized protein has a reduced affinity for its Mg^2+^ cofactor and is inactive. Reduced *Ps*cFP1 can be further inhibited by S-nitrosylation of Cys153. *Tb*FBPase has three cysteine residues but it is unclear whether this enzyme is similarly regulated. To date, no *Tb*FPBase post-translational modification has been reported on TritrypDB.

It is not surprising that metabolism changes with environment, cell density, and between different strains. These differences are often unpredictable and distinct processes respond differently. Variation in parasite responses provides insight into adaptive processes that are available to the parasites as they navigate different environments and highlights the importance of explicitly documenting growth conditions and cell lines. Further, the unexpected regulation of FBPase activity suggests an extra gluconeogenic function for FBPase. These studies indicate that our understanding of FBPase and its role in parasite biology is incomplete and worthy of further study.

## 4. Materials and Methods

### 4.1. Trypanosoma Brucei Cell Culture and Transfection

PF 2913-expressing T7 polymerase and tetracycline (tet) repressor [22] were maintained in either SDM79 [30], very low level glucose medium SDM79Θ (containing 5 μM glucose) [21], or SDM79ΘG (SDM79Θ supplemented with 5 mM glucose). For RNAi, nucleotides 262–780 of the FBPase open reading frame were cloned into pZJM [31] to generate pZJM:FBPase_262–780_. For transfection, 20 μg of plasmid DNA was linearized with *Not*I and electroporated in 4 mm cuvettes (BioRad GenePulser Xcell, Hercules, CA, USA; exponential, 1.5 kV, 25 μF). Twenty-four hours after transfection, culture was supplemented with appropriate drug for selection: 15 μg/mL G418; 50 μg/mL hygromycin; 2.5 μg/mL phleomycin. Cells were induced for 48 h with 1 μg/mL doxycycline daily before growth for assays. Cells were then harvested for both FBPase enzymatic assays and Western blot analysis. PF AnTat1.1 cell lines were generated by isolation of stumpy parasites after infection with long slender forms from a mouse, followed by differentiation to PF cells as described in [21]. These cells were then maintained in SDM79, SDM79Θ, or SDM79ΘG.

### 4.2. FBPase Enzyme Activity Assay

For FBPase assays, glycosome-enriched fractions were collected by differential centrifugation. Cells were harvested (1.3 × 10^8^, 800× *g*, 10 min), washed with PBS and mechanically lysed using 1 volume wet weight silicon carbide abrasive and filtered STE (250 mM sucrose, 25 mM Tris-HCl pH 7.4, 1 mM EDTA) with a protease inhibitor tablet (Thermo Fisher Scientific Pierce Protease Inhibitor Mini Tablets; Waltham, MA, USA). The abrasive was removed by centrifugation (100× *g*, 30 s) and supernatant was transferred to a new tube. The silicon carbide was washed an additional two times with filtered STE and the resulting supernatants were pooled. The collected supernatant was centrifuged (5000× *g*, 4 °C for 15 min) to remove nuclear debris and remaining intact cells. The supernatant was then transferred to a new tube and glycosomes and other small organelles were separated by centrifugation (17,000× *g*, 4 °C for 15 min), yielding a glycosome-enriched fraction. The supernatant was discarded and the glycosome-enriched pellet was resuspended in HBS-T (150 mM NaCl, 20 mM HEPES, 0.1% Triton). The master mix in sterile water (30 mM Tris-Cl, pH 7.2, 5 mM MgCl_2_, 0.5 mM NADP, 0.2 mM EDTA, 1 U/mL G6 PDH, 1 U/mL PGI, 2.5 mM fructose-1,6-bisphosphate (FBP)) was incubated for 20 min to allow impurities in F1,6BP to react with PGI and GAPDH. This step was included to reduce the background level of NADPH production. A total volume of 160 μL was added to each sample well. A cell equivalent of 4 × 10^7^ cell equivalents of glycosome-enriched pellet (40 μL) was added in triplicate to the plate. Recombinant *Tb*FBPase (1 μg) was used as a positive control, while a reaction mix lacking enzyme or lysate was used to score background. NADPH fluorescence was monitored spectrophotometrically on a Synergy H1 plate reader (excitation 340, emission 460) every thirty seconds for thirty minutes. Relative fluorescence units (RFU) per minute were calculated based on the slope of the linear portion of the reaction (10–20 min). For cell density-dependent assays, cells were seeded at 5 × 10^5^ cells/mL and grown to a log phase density of 5 × 10^6^ cells/mL and a stationary phase density of 10^7^ cells/mL without passage.

### 4.3. Western Blot Analysis

Cells were harvested (5 × 10^6^, 800× *g*, 10 min), washed with PBS, lysed in cracking buffer (10% glycerol, 2% SDS, 2% *β*-mercaptoethanol, 100 mM Tis (pH 6.8), 0.1% bromophenol blue), and boiled at 95 °C for 3 min. Proteins were resolved in the lysate by 12% SDS-PAGE, followed by transfer to nitrocellulose. Membranes were then probed with antibodies against *Tb*FBPase (1:10,000; the generous give of Prof. Fred Bringaud, (Université de Bordeaux, Bordeaux, France), tubulin (1:20,000), and aldolase (1:10,000).

## Figures and Tables

**Figure 1 pathogens-10-00617-f001:**
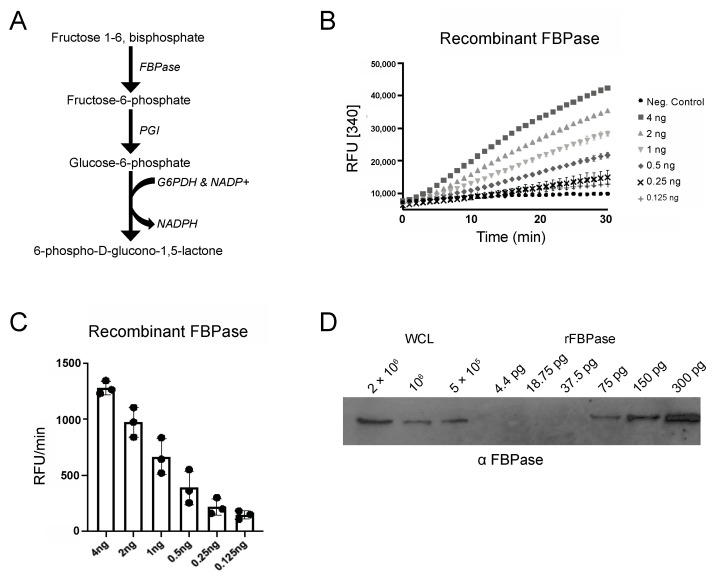
FBPase assays sensitivity. (**A**) FBPase activity was measured in a coupled-enzyme assay containing phosphoglucoisomerase (PGI) and glucose-6-phosphate dehydrogenase (G6PDH). The conversion of NADP+ to NADPH was measured via fluorescence at 460 nm after excitation at 340 nm [340]. (**B**) Serial dilutions of recombinant FBPase (rFBPase, 125 pg–4 ng) were used in FBPase assays and the relative fluorescence units [RFU] per minute plotted as a function of time. (**C**) Bar graph shows RFU/min for three technical replicates. (**D**) Quantification of FBPase in parasite lysate. Serial dilutions of rFBPase and whole cell lysate (WCL) were analyzed by Western blotting with anti-TbFBPase antibodies.

**Figure 2 pathogens-10-00617-f002:**
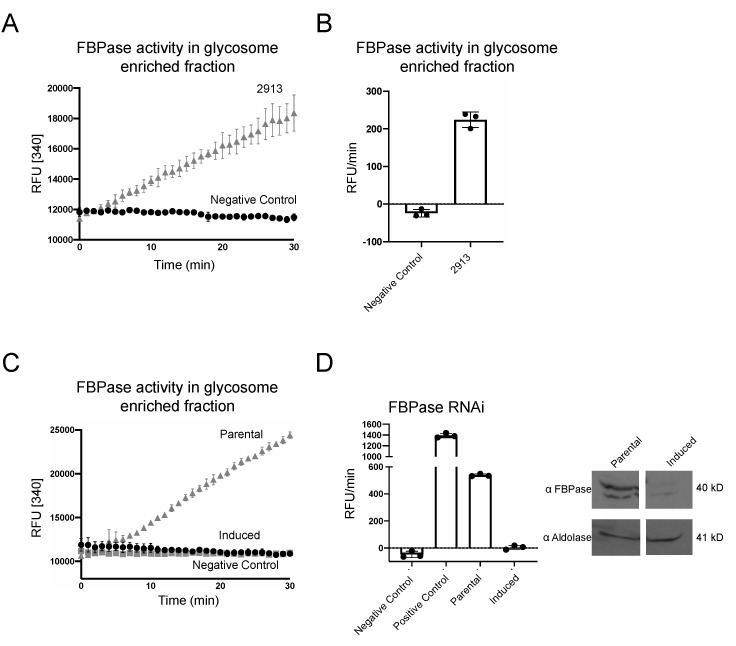
PF 2913 cells grown in SDM79 have FBPase activity. (**A**) Reaction curve of FBPase activity in glycosome-enriched fractions of PF 2913 (grey triangles) and negative control (no lysate, black circles). (**B**) Bar graph of FBPase activity. (**C**) FBPase activity is undetectable in FBPase RNAi cell lines. Reaction curve of FBPase activity in parental PF 2913 (grey triangles), induced RNAi cell lines (grey squares) and negative control (black circles). (**D**) Bar graph of FBPase activity (RFU/min) showing data from one representative biological replicate performed in technical triplicate. Inset, Western blot of parental 2913 PF and induced FBPase RNAi cell lysates probed with antibodies against FBPase and aldolase, which was used as a loading control.

**Figure 3 pathogens-10-00617-f003:**
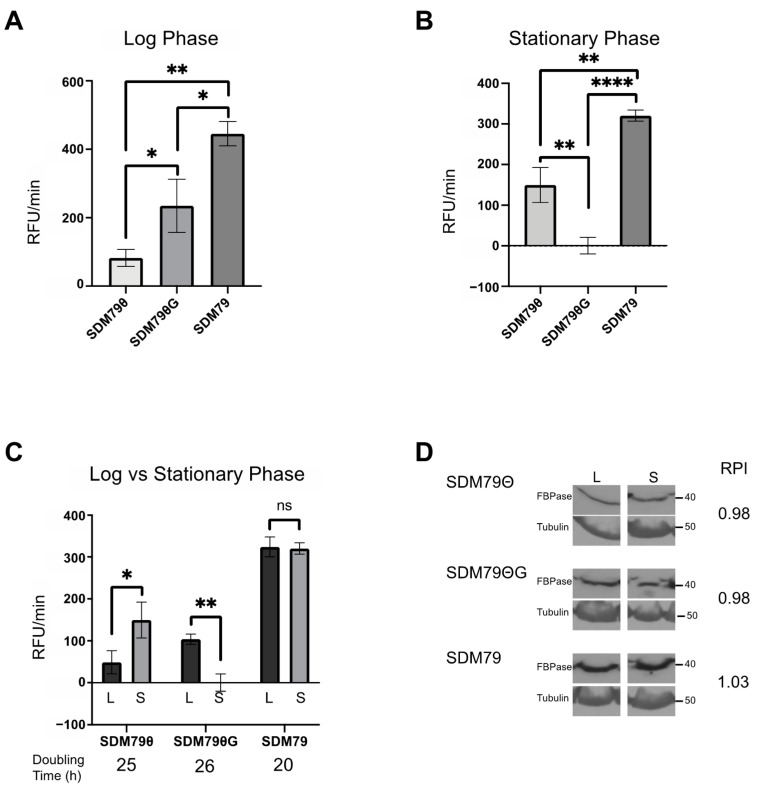
FBPase activity is influenced by extracellular glucose and cell density. FBPase activity was measured in PF 2913 cells grown in SDM79Θ, SDM79ΘG, and SDM79. (**A**) Bar graph of FBPase activity in cells grown to log phase (5 × 10^6^/mL) in each of the described media conditions. (**B**) Bar graph of FBPase activity in cells grown to stationary phase (1 × 10^7^/mL) in each of the described media. (**C**) Bar graph of FBPase activity comparing same media. Log (black bars) vs. stationary phase (grey bars). For all assays, FBPase activity was calculated using three biological replicates performed in triplicate. (**D**) Western blots of whole cell lysates (log phase, L and stationary phase, S) probed with anti-FBPase and anti-Tubulin used as a loading control. Pixel intensities of FBPase were calculated in FIJI and normalized to Tubulin. Ratios of log phase FBPase pixel intensity to stationary phase FBPase pixel intensity are given as Relative Pixel intensity (RPI). Error bars represent standard deviation. Significance was determined using Student’s *t*-tests of sample pairs indicated by brackets (* *p* < 0.05, ** *p* < 0.01, **** *p* < 0.0001).

**Figure 4 pathogens-10-00617-f004:**
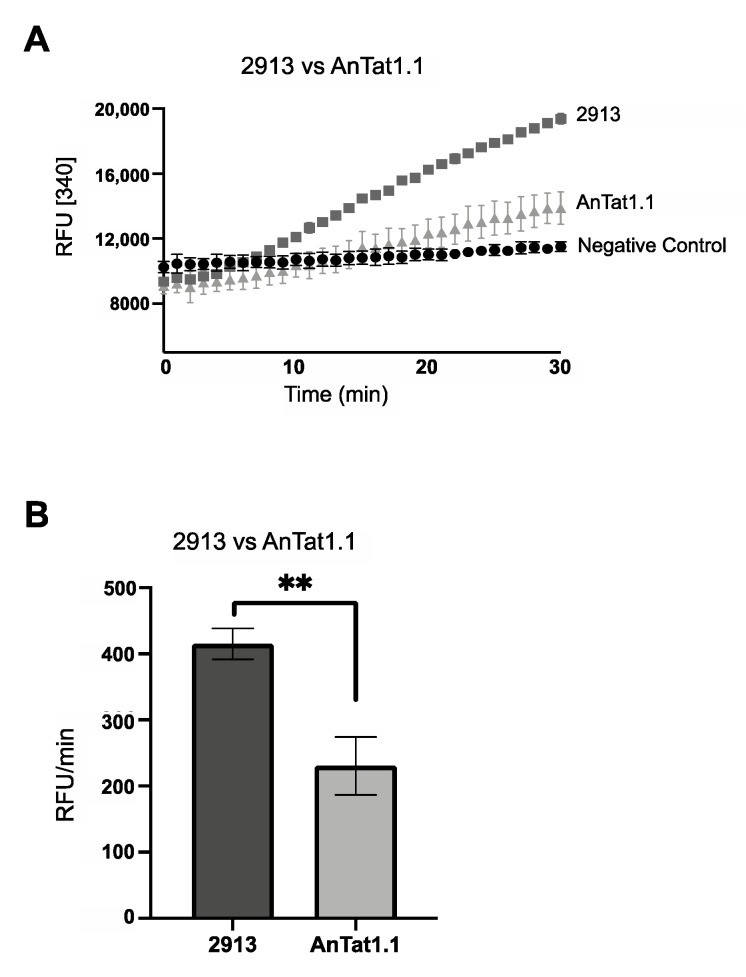
FBPase activity is higher in PF 2913 as compared to PF AnTat1.1 when both are grown in SDM79. (**A**) FBPase activity in PF 2913 (dark grey squares), PF AnTat1.1 (light grey triangles) and negative control (black circles). (**B**) Bar graph of FBPase activity (RFU/min) of representative biological replicate performed in triplicate (** *p* < 0.01, Student’s *t*-test). PF 2913 (black bar) and PF AnTat1.1 (grey bar).

**Figure 5 pathogens-10-00617-f005:**
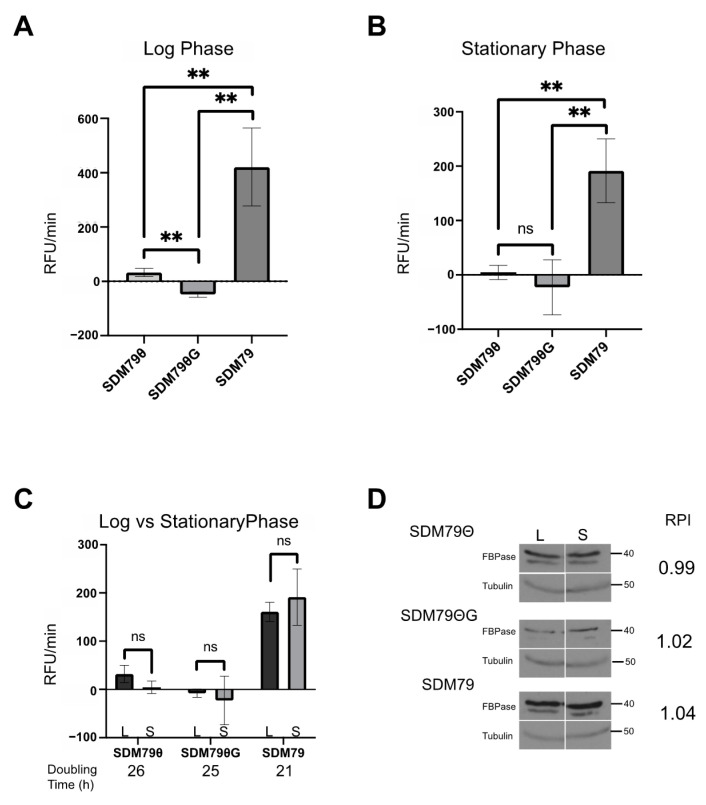
In PF AnTat1.1 cells, FBPase activity was undetectable in cells grown in SDM79Θ and SDM79ΘG but present in SDM79. FBPase activity was measured in PF AnTat1.1 cells grown in SDM79Θ, SDM79ΘG, and SDM79. (**A**) Bar graph of FBPase activity in cells grown to log phase (5 × 10^6^/mL) in each of the described media conditions. (**B**) Bar graph of FBPase activity in cells grown to stationary phase (1 × 10^7^/mL) in each of the described media. (**C**) Bar graph of FBPase activity comparing the same media log (black bars) vs. stationary phase (grey bars). All assays were performed with three biological replicates in triplicate (log phase, L and stationary phase, S). (**D**) Western blots of whole cell lysates (log phase, L and stationary phase, S) were probed with anti-FBPase and anti-Tubulin. Significance was determined using Student’s *t*-tests of sample pairs indicated by brackets (** *p* < 0.01).

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
