# Peer review of "Regulation of Fructose 1,6-Bisphosphatase in Procyclic Form *Trypanosoma brucei"

_pathogens, 2021, doi:10.3390/pathogens10050617_

Round 1
Reviewer 1 Report
It has long been known that the key gluconeogenic enzyme fructose-1,6-bisphosphatase (FBPase) is present in procyclic Trypanosoma brucei but until recently the functional role of this enzyme had been unstudied. Work from another group had shown that whilst the gene encoding FBPase is non-essential in culture, it is required for natural fly transmission of T. brucei. It was also known that fructose-6-phosphate production in a gluconeogenic direction in the absence of FBPase is possible. Here, Wilkinson and Morris report data from a series of biochemical studies investigating FBPase activity in cultured parasites where the parameters varied were (i) nature of the media used; (ii) stage of a growth curve (exponential versus stationary phases); and (iii) different T. brucei strains. Whilst it’s difficult to make sense of data that as the authors acknowledge often seems counter-intuitive, the data are convincing and this is an intriguing dataset that will interest those working in the molecular parasitology field, most obviously those working on trypanosomatid metabolism. There is also a well-considered (in my view) speculation on the role that FBPase activity in glucose-replete conditions may have for regulation of pyruvate kinase.
Revisions requested for a short, straightforward manuscript:
Lines 108-138 and Figure 3. I accept and follow how the cells were analysed at different point of a batch-fed growth cycle, but the missing piece of data is the statement of the generation (or cell-doubling) time of the cultures analysed.
Lines 155-158. How do PF Antats differentiate in culture? Do they differentiate through mesocyclic, epimastigote and thence metacyclic forms in tissue culture? I didn’t think this was the case but BF Antats readily differentiate BF-to-stumpy-to-PF. The authors simply need to be clearer in their writing here, so as to remove ambiguity.
Lines 271-277. The authors (appropriately) make comparison with biology in Leishmania and Pisum. However, I recommend that speculation is best made by noting that the former is a distantly related trypanosomatid parasite and the latter is a plant.
Reviewer 2 Report
The manuscript entitled "Regulation of fructose 1,6-bisphosphatase in procyclic form Trypanosoma brucei" revealed the enzymatic activity of FBPase and suggested that the enzyme was regulated by cell density. This is the first report for the presence and activity of the enzyme in PF. The authors revealed the enzymatic activity by enzymatically method and RNAi. The methodology is robust and the results looks fine. However, some of improvement points are remained.
- Fig.1-D: Western blot
Between 5x10^5 and 300 pg, the reviewer found line. If you analyzed WCL and rFBPase separately, the authors must indicate line between the photos.
- Fig.2 and other figs.
Some of error bars were shown in figures. The author have to clarify the meaning of them, standard deviation or standard error.
In addition, the author should modify the barographs with error bar (e.g. Fig.2-B) to identify each points like as Fig. 2-D.
- Statistical analysis among SDM0, SDM799G and SDM79.
The author applied "t-test" for statistical analysis among the group. It is not acceptable because of increasing α-type error. The author must analyze them by ANOVA and post-hoc test (maybe Tukey-Kramer's test is applicable).
- Western blot analysis for FBPase
The author discuss the difference of the amount of FBPase based on western blot analysis: However, the author did not shown the quantitative analysis for them. The author should re-analyze them if the author will emphasize the post-translational regulation of FBPase. In addition, the western blots are not so clear because each signal are leaning and combined with other blots.
- Cultured trypanosomes 2931 and AnTat1.1
The author mentioned the historical difference of them in the manuscript. How is the their biological difference in these medium? Were their no difference of doubling time etc. among strains and medium conditions? You should show the growth curve of them, at least. Yes, I agree the FBPase activity might be caused the difference of their history. But I also think their biological performance in the culture might affect the result.
Reviewer 3 Report
Authors in this manuscript measured the activity of fructose bisphosphatase (FBPase) in Trypanosoma brucei prpcyclic form (PF) grown in the presence and absence of glucose and at different stages of cell growth. FBPase is a key gluconeogenic enzyme, localized in glycosomes in T. brucei. However, its role and regulation are controversial in this organism that is partly due to difficulties to measure the activity of this enzyme in cell lysate. Authors used a glycosome enriched fraction of T. brucei and showed activity of FBPase by means of a sensitive fluorometric assay. Results described that FBPase activity varies with culture conditions and in different strains but in somewhat unexpected ways. FBPase activity was higher when cells were grown in glucose-containing than in glucose-depleted media, where gluconeogenesis should be increased. Authors also demonstrated that FBPase activity varies widely between a laboratory adapted monomorphic and a freshly differentiated pleomorphic PF strains. To explain their unexpected results authors mentioned that FBPase could have other roles besides gluconeogenesis or FBPase may not be the key enzyme and other enzyme activities in pentose phosphate pathway are needed to increase gluconeogenesis. To authenticate their assay, authors used glycosome-enriched fraction isolated from FBPase knockdown T. brucei as a negative control. Overall, the studies are interesting, and authors demonstrated the way to measure the FBPase activity in T. brucei. One concern is that authors didn’t test the glycosome prep (without addition of any substrate) as a blank. This could eliminate the possibility of contamination of endogenous substrate, NADPH, or interference of other enzymes. Additional minor points are given below.
- 3. A and B could be removed, since Fig. 3C is showing the same result in a combined way
- Same for Fig. 5. A and B could be removed, since Fig. 5C is showing the same result in a combined way
- Line 77; ‘recombinant protein FBPase protein’—remove the first protein
- Line 127; ‘medias’ remove s
- Line 150, PF 2913s, remove s
- Line 212; ‘medias’ remove s
- Line 294; SDM79(-) should have 5 ?M glucose
Round 2
Reviewer 3 Report
Authors mentioned that some supporting data has been provided in the supplemental file. However, the reviewer didn't see any supplemental files attached
Fig. 1D. A line need to be shown in between the lane 3x105 and 300 pg
Authors responded to other concerns adequately
Author Response
We apologize to reviewer 3. We had mistakenly uploaded a version without updated figure 1 and supplemental figure. We have uploaded the corrected version.